# The Construction and Evaluation of a Multi-Task Convolutional Neural Network for a Cone-Beam Computed-Tomography-Based Assessment of Implant Stability

**DOI:** 10.3390/diagnostics12112673

**Published:** 2022-11-03

**Authors:** Zelun Huang, Haoran Zheng, Junqiang Huang, Yang Yang, Yupeng Wu, Linhu Ge, Liping Wang

**Affiliations:** 1Guangzhou Key Laboratory of Basic and Applied Research of Oral Regenerative Medicine, Guangdong Engineering Research Center of Oral Restoration and Reconstruction, Affiliated Stomatology Hospital of Guangzhou Medical University, Guangzhou 510182, China; 2Department of Chemical & Materials Engineering, University of Auckland, Auckland 1010, New Zealand; 3Department of Stomatology, The Third Affiliated Hospital of Guangzhou Medical University, Guangzhou 510150, China

**Keywords:** dental implants, implant stability, CBCT, ISQ, deep learning, cascade network

## Abstract

**Objectives:** Assessing implant stability is integral to dental implant therapy. This study aimed to construct a multi-task cascade convolution neural network to evaluate implant stability using cone-beam computed tomography (CBCT). **Methods:** A dataset of 779 implant coronal section images was obtained from CBCT scans, and matching clinical information was used for the training and test datasets. We developed a multi-task cascade network based on CBCT to assess implant stability. We used the MobilenetV2-DeeplabV3+ semantic segmentation network, combined with an image processing algorithm in conjunction with prior knowledge, to generate the volume of interest (VOI) that was eventually used for the ResNet-50 classification of implant stability. The performance of the multitask cascade network was evaluated in a test set by comparing the implant stability quotient (ISQ), measured using an Osstell device. **Results:** The cascade network established in this study showed good prediction performance for implant stability classification. The binary, ternary, and quaternary ISQ classification test set accuracies were 96.13%, 95.33%, and 92.90%, with mean precisions of 96.20%, 95.33%, and 93.71%, respectively. In addition, this cascade network evaluated each implant’s stability in only 3.76 s, indicating high efficiency. **Conclusions:** To our knowledge, this is the first study to present a CBCT-based deep learning approach CBCT to assess implant stability. The multi-task cascade network accomplishes a series of tasks related to implant denture segmentation, VOI extraction, and implant stability classification, and has good concordance with the ISQ.

## 1. Introduction

Implant stability refers to the healing of a dental implant [1], and the success of implant restoration directly depends on the clinician, the control of implant stability, and the prediction of changes in implant stability [2]. In clinical treatment, implant stability is often assessed using resonance frequency analysis (RFA) [3], which can be converted into an implant stability quotient (ISQ), with a higher ISQ value indicating higher implant stability and more extensive osseointegration [4]. Generally, the ISQ of dental implants ranges from 40 to 80 [5]. ISQ measurement requires twisting a Smartpeg into the implant and measuring it with an Osstell device, which is not possible during submerged healing.

Imaging techniques are widely used to assess the mass of alveolar bones, but there are many limitations to using conventional periapical radiographs and panoramic radiographs when assessing implant stability, as they do not provide information on the buccolingual alveolar bone, and bone loss on the buccolingual side of the implant precedes the mesial-distal side [6]. Assessment of the peri-implant osseointegration region (the bone–implant interface) using cone-beam computed tomography (CBCT) can be performed to effectively assess implant stability [7]. However, it requires the assistance of a specialized radiologist, and there is the possibility of errors in individual subjective judgments. Artificial intelligence may be an effective solution, especially convolutional neural networks (CNNs), which are specialized artificial neural networks that use convolution in at least one layer instead of matrix multiplication. CNNs are mainly used to process pixel data and in image recognition and processing [8]. The main applications of CNNs to CBCT images are semantic segmentation [9], feature extraction [10], and diagnosis [11].

With the development of deep learning, several CBCT image segmentation methods have been developed. For example, the MS-D-based CNN can be used for the classification identification and segmentation of teeth and jaws in CBCT [12]. Liu et al. [13] achieved detection and pixel-level detection of mandibular third molars and mandibular neural canals based on U-net segmentation and classification of the M3–MC relationship using ResNet-34. In addition, deep learning for disease diagnosis using CBCT has shown excellent results [11]. Some scholars have combined pre-trained DenseNet models with pathological information to construct a deep learning method for detecting transmissive lesions of jawbones through CBCT images [14].

To the best of our knowledge, the usefulness of CNN for the prediction of implant stability remains to be explored, having been explored in no prior studies. This study combined prior knowledge with a cascade network and proposed a deep learning model to assess implant stability for the first time. First, we had the multi-task cascade network perform implant denture segmentation using the MobilenetV2-DeeplabV3+ model; this was combined with an image processing algorithm in conjunction with osseointegration theories [15] to generate a volume of interest (VOI) to feed to the ResNet-50 model for implant stability classification. Through this study, we hope to validate the potential for CNNs to guide and provide direction for dental implant therapy.

## 2. Materials and Methods

### 2.1. Surgical Protocol and Resonance Frequency Analysis 

This study was conducted in accordance with the Declaration of Helsinki and recent TRIPOD guidelines for prediction model development and validation. This study was approved by the Ethics Committee of the Third Affiliated Hospital of Guangzhou Medical University (2022-093). All participants signed an informed consent form prior to inclusion. Clinical data, including CBCT images and ISQ values for the same period, were acquired from July 2021 to June 2022 from 41 patients who requested dental implant therapy at the Department of Stomatology of the Third Affiliated Hospital of Guangzhou Medical University, including 23 males and 18 females (for a total of 55 implants). The inclusion criteria were as follows: (1) age 18 years or older, (2) patients with indications for implants, and (3) provision of written informed consent. The exclusion criteria were as follows: (1) severe periodontal disease; (2) the need for complex bone augmentation surgery; (3) possible severe distortion or scattered artifacts in the CBCT images due to orthodontic treatment or metal restorations; and (4) maxillofacial contraindications to implant surgery, such as a history of radiation therapy, severe diabetes, or hypertension.

All implants were placed using the conventional implant surgery protocol at least 6 weeks after extraction, and implants of the Strauman Dental Implant System were placed. CBCT was performed immediately and 3 months after implant placement, and measurements were taken using an Osstell device (Osstell AB, Sampgatan, Goteborg, Sweden) to measure ISQ values. Buccal lingual and proximal–distal mesial ISQ values were measured, and the mean values were recorded to ensure reproducible measurements.

### 2.2. CBCT Image Acquisition and Image Pre-Processing

CBCT was performed using NewTomVG (FSV: 90 kV; mAs: 12.92; CTDIvoi: 1.61 mGy; exposure time: 3.6 s), and the data were stored in DICOM format. All personally identifiable information was removed using anonymization. The CBCT data were imported into NewTom NNT 9.0.0 and reconstructed according to the sagittal arch, with a thickness of 0.3 mm to cross-sectional images vertical to the arch. In total, 20,488 cross-sectional images of the arch and 779 implant images were obtained.

For the 779 coronal section implant images, an experienced radiologist used the data annotation software Labelme (Windows 5.0.1, MIT, Cambridge, MA, USA) to annotate the images containing implant dentures (Appendix A).

### 2.3. Estimation of Sample Size

To determine the test set sample size, we calculated the sample size using the single-arm diagnostic trial sample estimation formula in PASS 15 (Power Analysis and Sample Size Software, 2017; NCSS, LLC, Kaysville, Utah, USA, ncss.com/software/pass). Based on the training set results, a sensitivity of 95%, a specificity of 88%, and an α of 0.05, the test set required at least 133 images. We randomly selected 20% of the entire dataset as the test set, and the final sample size was 155 images each for the binary and quaternary classification test sets and 150 images for the ternary classification test set.

### 2.4. Construction of a Multi-Task Cascade Network

The cascade network (Figure 1) constructed in this study consists of the following modules: (A) a MobilenetV2-DeeplabV3+ implant recognition and segmentation network, (B) a VOI extractor combined with prior knowledge, and (C) a ResNet-50 implant stability classification network. In the test set, cross-sectional images of the entire dental arch were fed into the trained MobilenetV2-DeeplabV3+ network; and only the implant cross-sectional images in which the implants were segmented were output. Based on the semantic segmentation results, the image of the peri-implant osseointegration region as VOI was obtained by combining implant information (e.g., implant length and tooth position). The VOI was finally fed into the trained ResNet-50 network, and the ISQ classification results were output. Finally, we observed the accuracy and efficiency of the cascade network in the binary, ternary, and quaternary ISQ classification test sets.

#### 2.4.1. Construction of the MobilenetV2-DeeplabV3+ Implant Recognition and Segmentation Network

Most semantic segmentation networks comprise encoding and decoding blocks. The encoding block progressively downsamples the input image, whereas the resolution of the feature map is reduced step-by-step to capture deeper feature information. The decoding block is used for upsampling the small-sized feature map to generate a segmentation result with the exact resolution of the original image. DeeplabV3+ uses the same “encoding block, decoding block” construction, using a modified Xception network [16], applying depth-separated convolution [17], and incorporating batch normalization(BN) [18] and ReLU [19] to improve feature extraction.

In the encoding block, atrous spatial pyramid pooling (ASPP), which performs parallel cavity convolution (atrous convolution) on the initial effective feature layers that are compressed four times, is used for feature extraction by parallel convolution with different ratios, followed by concat merging, and finally compressing the features using 1 × 1 convolution. In the decoding block, low-level features underwent 1 × 1 convolution to reduce the number of channels. High-level features were upsampled four times bilinearly and then linked to the corresponding low-level features. The features are linked or refined by the feature layers by 3 × 3 convolution, and quadruple bilinear upsampling is performed.

In this study, we improved the above DeeplabV3+ network at the network level as follows: at the backbone network level, we replace the original version of the Xception backbone network to MobilenetV2 [20], considering the potential need for this study to operate in embedded devices. The entire MobilenetV2 consists of an inverted residual block as the core component; the inverted residual block network structure is spindle-shaped, first up-dimensioned by 1 × 1 convolution, then by deep separable convolution, and finally down-dimensioned. The residual edges for the tensor superposition link the inputs and outputs of each block. MobilenetV2 performed a total of four down-samplings during the feature extraction process. The feature maps generated after the first two down-samplings were completed were further used as low-level features, while those generated after all down-samplings were completed were used as high-level features. The ASPP part is consistent with the decoder part and DeeplabV3+ (Figure 2).

#### 2.4.2. Training of the MobilenetV2-DeeplabV3+-Based Implant Recognition and Segmentation Network

To prevent model overfitting and positive and negative sample imbalances, online augmentation was used to expand the dataset by panning, rotating, and flipping each batch of data. Moreover, we used focal loss to solve the problem of a serious imbalance in the ratio of positive and negative samples in one-stage target detection, which reduces the loss function to reduce the weight of negative samples during training. Meanwhile, we loaded MobilenetV2 pre-training weights, which were trained using the ImageNet dataset [21].

The MobilenetV2-DeeplabV3+ was trained for 50 epochs. The specific implementation and associated parameters used in the training process were as follows: batch size of 16, initial learning rate of 5 × 10^−4^, and an input image resolution of 256 × 256. To accelerate the convergence of the network in the expected direction and avoid oscillations at the nadir attachment, we also used the stepLR learning rate decay function with a gamma of 0.94. The learning rate decreases once for each completed epoch of training. We used Adam as an optimizer to feature a fast convergence at the beginning of training. The downsampling multiplier for the backbone network was 8. The final training used cross-entropy loss + dice loss as the loss function, because dice loss exhibits good performance for scenarios with a severe imbalance between positive and negative samples, and the training process focuses more on the mining of foreground regions.

#### 2.4.3. Prior Knowledge-Based VOI Extractor

The output of the MobilenetV2-DeeplabV3+ semantic segmentation network is the implant denture segmentation image, which is also used as the input of the VOI extractor. The processing flow of the VOI extractor is as follows. First, the pixels belonging to the implant are hidden according to the segmentation result. Since the bone mass around the implant is highly correlated with implant stability [22], we obtained the implant and surrounding bone images by extending the segmented edge horizontally by 5 pixels. According to the tooth position, all implant images placed in the mandible were flipped so that all implant apexes were on top, and the implant neck and abutment faced downward. Subsequently, the implant and peri-implant tissue images were extracted by cropping them according to the implant length. Finally, the images of the peri-implant osseointegration region were obtained and used as the VOI feed to the next stage.

#### 2.4.4. Construction of the ResNet-50-Based Implant Stability Classification Network

To overcome the problem of low learning efficiency and accuracy degradation, we used ResNet-50 [23] as a classification neural network. ResNet directly introduces a layer of data from several previous layers into the input part of the later data layers by skipping multiple layers. One of the preceding layer lines partially contributes to the content of the last feature layer. The benefit of residual connection is that it allows the raw information from the lower levels to be passed directly to the subsequent higher levels, allowing the higher levels to focus on residual learning and avoiding accuracy degradation (Appendix A).

ResNet-50 has two modules: Conv Block and Identity Block, where the Conv Block input and output dimensions are different, so they cannot be connected in successive series, and their role is to change the dimension of the network. The identity block input and output dimensions were the same, and could be connected in series to deepen the network (Appendix A). For the different classification modes, the number of neurons in the last fully connected layer was set to category (Figure 3). 

#### 2.4.5. Training of the ResNet-50-Based Implant Stability Classification Network

In this study, the 779 VOI images generated in the VOI extractor section were used as the dataset for the classification neural network. We designed three ISQ classification modes. The first is a binary classification with ISQ 65 demarcations. The second is a ternary classification with ISQ 60, 70 as the demarcation, and the third is a quaternary classification with ISQ 50, 60, 70 as the demarcation. The dataset was replicated using three different ISQ classification patterns. Each replica was then randomly divided into training and test sets with a ratio of 8:2. Random data augmentation and freeze/unfreezing of the model structure were used to perform training.

The training process was based on a pre-trained model, and the first 50 epochs were trained with a frozen backbone, which was then unfreezed to train another 950 epochs. The specific implementation and related parameters used in the training process were as follows: an input image resolution of 64 × 64, a cosine dynamic learning rate, an initial learning rate of 1 × 10^−3^, a minimum learning rate of 1 × 10^−5^, and an Adam optimizer. ResNet-50 uses cross-entropy as the loss function.

### 2.5. Model Performance Evaluation and Statistical Analysis

Two deep learning methods were used in this cascade network: implant segmentation and ISQ classification. The following validation criteria were used to verify the model’s performance. The pixel accuracy (PA), mean pixel accuracy (mPA), and intersection over union (mIoU) were used to evaluate the semantic segmentation model. The performance of different ISQ classification models was evaluated by recall, accuracy, and confusion matrices; based on the confusion matrix, the multi-classification was converted into binary classification to evaluate the sensitivity, specificity, accuracy, PPV, NPV, and F1 scores between groups in the model. Statistical analysis and data visualization were performed using R (version 3.6.3) and the R package ggplot2 (version 3.3.3).

## 3. Results

### 3.1. Performance of Implant Identification and Segmentation Based on MobilenetV2-DeeplabV3+

Table 1 shows the performance of MobilenetV2-DeeplabV3+ applied to the edge segmentation of implant dentures. After evaluating the test set, the MobilenetV2-DeeplabV3+ network exhibited good segmentation performance, with mIoU achieving 94.4%, mPA 96.76%, average recall of 96.87%, average precision of 97.33%, and accuracy of 99.76%.

### 3.2. Classification Performance of Implant Stability Based on ResNet-50

The trained ResNet-50 model was tested on different classification test sets. We found that 149 out of 155 images were correctly classified in the binary classification test set, with an accuracy of 96.13%, a precision of 96.20%, and a mean recall (mRecall) of 96.06%. The ternary classification test set had 150 images, and 143 images were correctly predicted with 95.33% accuracy, 95.33% precision, and 95.37% mRecall. The quaternary classification test set had 155 images, and 141 images were correctly predicted with 92.90% accuracy, 93.71% precision, and 93.01% mRecall. 

The prediction results of each classification test set are shown in Figure 4a–c. The horizontal coordinates show different classification test sets. The vertical coordinates represent the actual ISQ value. The box plots the distribution interval of the positive predicted values in each test set, the median, quartiles, and other indicators, and the violin plot depicts the distribution status of the positive predicted values and numerical density. We used the points on the upper or lower edges of the box plot to represent the negative predictive values.

We produced confusion matrices (Figure 4d–f) based on the classification results. We converted the multi-classification into binary classification to retrieve various metrics, including the accuracy, sensitivity, specificity, PPV, NPV, and F1 scores, as shown in Table 2.

### 3.3. Time Costs

The entire implant stability evaluation process is shown in Figure 1, and the entire cascade network was subjected to time-consuming tests. The test platform was an Intel Core i5-11400F CPU and NVIDIA RTX3060 GPU. We randomly selected 20 implants from the dataset, performed ISQ prediction on their cross-sectional images, and recorded the time taken to complete all ISQ predictions for each implant. The above procedure was performed for 10 rounds, and the results showed that ISQ prediction of all cross-sectional images of one implant was completed every 3.76 s, on average.

## 4. Discussion

Implant stability is a process that dynamically changes with bone remodeling after implant placement. Using medical imaging techniques to assess the peri-implant bone can assist in the assessment of implant stability. For example, fractal analysis can be used to assess peri-implant bone tissue healing and trabecular bone structure using ultrasound [24]. It can also be applied to panoramic films to assess implant stability [7]. In this study, we proposed—for the first time—a multitasking cascade network based on CBCT to assess implant stability by identifying and segmenting implants in whole dental arch cross-sectional images using the MobilenetV2-DeeplabV3+ network. We then combined osseointegration theory and segmentation results to extract VOI on cross-sectional images. Finally, the depth residual network Resnet-50 was used for ISQ classification. The evaluation results in the test set show that Resnet-50 performs well in the task of ISQ classification.

The ISQ is a globally standardized method for calculating implant stability. According to the literature, ISQ values greater than 70 are considered as having high stability, 60 to 69 are considered as having moderate stability, and values below 60 are considered as having low stability [25,26,27]. The ITI International Consensus provided clinical recommendations for implant loading protocols. Specifically, the suggestions were that the ISQ should be greater than 60–65 for single implant loading and greater than 60 for implant ISQ for overdenture restorations in edentulous patients [28]. A prerequisite for the immediate restoration of a single implant is a primary stability ISQ greater than 65 [29], and for immediate loading implants, an ISQ greater than 70 is recommended [30]. Based on the clinical significance of implant stability, to evaluate the network’s diagnostic performance, the output was classified using a binary classification with an ISQ of 65 as the cut-off, a ternary classification with ISQ cut-offs of 60 and 70, and a quaternary classification with ISQ cut-offs of 50, 60, and 70.

The training of deep learning models is carried out through a large number of image and label correspondences, with no human intervention during this period. However, deep learning in medical image processing by introducing prior knowledge often achieves better performance and has higher interpretability. For instance, Zheng et al. [31] introduced an anatomical knowledge-based score function into the U-Net’s objective function using CBCT images to achieve good performance. Moreover, Li et al. [32] started from a priori knowledge, using the retina’s structural properties around the optic nerve papillae, and employed neural graph networks to capture the spatial structural relationships of medical images. Moreover, a two-level segmentation network framework, assisted by multiscale graph networks, was designed to improve the segmentation performance of deep learning networks [32].

The influence of peri-implant bone on implant stability is well known, and can be inferred from peri-implant bone density, volume, and cortical thickness on CBCT [33]. To avoid the noise caused by redundant image information that interferes with the resultant classification network, it is necessary to remove the background, the implant superstructure, and the bone tissue on the buccolingual side away from the implant, because these are not relevant to implant stability. Accordingly, we extracted additional feature information manually and excluded the effects of noise.

The diagnosis of medical images is essentially a computer vision problem, but it often has many unique industry-specific issues. The lack of data is one of the factors limiting the development of deep learning for medical-aided diagnoses. For such problems, the migration of a trained model on a task (source domain) to another task (target domain) [34] is a good solution. In this study, we modified the original deeplabV3+ architecture by changing its original backbone to MobilenetV2, freezing the backbone of the pre-trained model loaded with ImageNet during the training process, and training only the layers outside the backbone. Out of the total number of parameters of approximately 580 W, approximately 181 W parameters were untrained. In addition, in the ResNet-50 implant stability classification network, we loaded the pre-trained model of ImageNet. We then froze the backbone for the first 50 epochs of training, and then unfroze the backbone for the other 950 epochs. Eventually, the model achieved satisfactory accuracy.

End-to-end means that the parameters that need to be determined in the original steps are learned jointly, rather than in steps. The key to end-to-end learning is whether there is sufficient data to directly learn complex functions to map from x to y [35]. The most significant advantages of end-to-end learning are its simplicity and the fact that it does not require a complex process to be designed [36]. In contrast, in end-to-end models, it is more difficult to determine how the components of the model contribute to the end goal, which leads to a reduction in the interpretability of the overall architecture. In addition, it is more challenging to incorporate a priori knowledge into the end-to-end model, which prevents us from providing direction or limiting the learning of the entire network based on clinical experience.

The idea of multitask learning provides an alternative solution to our research. Multitask learning can learn multiple tasks simultaneously by sharing models, and can improve data efficiency, reduce overfitting by sharing representations, and perform fast learning by using auxiliary information [37]. We refer to the design idea of multi-task cascade networks, where the output of each task-specific branch is appended to the following task-specific input, forming a “cascade” of information flow by combining MobilenetV2-DeeplabV3+-based implant recognition and segmentation. The automated prediction of implant stability was achieved by cascading the MobilenetV2-DeeplabV3+ implant identification and segmentation network, the VOI extractor with prior knowledge, and the ResNet-50 network.

Although high accuracy was obtained for the test set, there are still some limitations. First, we only used one brand of implant, and different implant systems have different implant designs and surface treatments, which have different effects on implant stability. Moreover, because the deep learning method hinders its interpretability, medical diagnostic systems must be transparent and interpretable to gain the trust of doctors and patients. In future research studies, additional datasets from different implant systems will be collected to improve the generalization of our multitask cascade network, and to develop better visualization efforts to improve its interpretability. Second, the current diagnostic model only assesses immediate implant stability. We aimed to predict changes in implant stability over time by tracking the relationship between changes in CBCT images and implant stability during implant bone healing.

## 5. Conclusions

This study represents a successful first step towards showing that that deep learning methods may be potentially useful for implant stability measurement. In addition, we were able to construct a cascade network that had good concordance with the ISQ value measured using the Osstell device. 

## Figures and Tables

**Figure 1 diagnostics-12-02673-f001:**
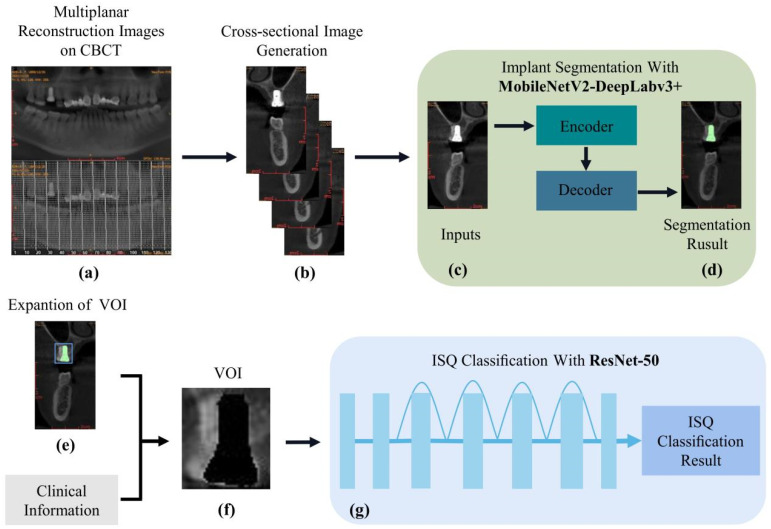
(**a**) Workflow of the multi-task cascade network; (**b**) CBCT image reconstruction; (**c**,**d**) cross-sectional image input to the MobilenetV2-DeeplabV3+ network and output implant dentures segmentation image. The green section represents the implant dentures; (**e**,**f**) a VOI extractor, combined segmentation results with clinical information extensions to obtain the VOI image; (**g**) importation of the VOI into the ResNet-50 network to output classification results. CBCT, cone-beam computed tomography.

**Figure 2 diagnostics-12-02673-f002:**
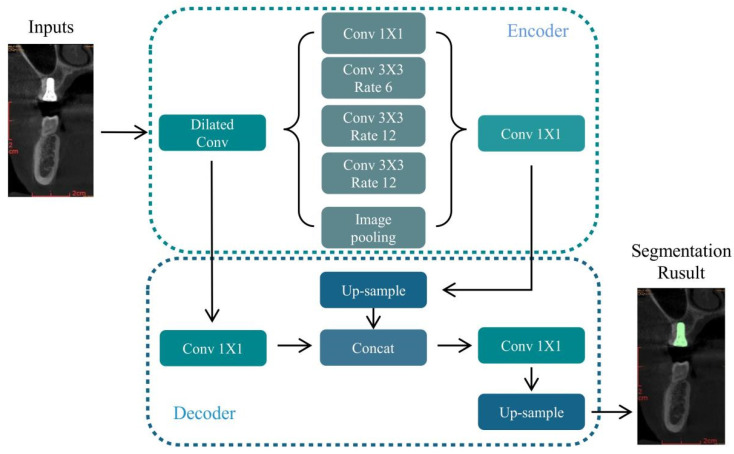
Network architecture of MobilenetV2-DeeplabV3+.

**Figure 3 diagnostics-12-02673-f003:**
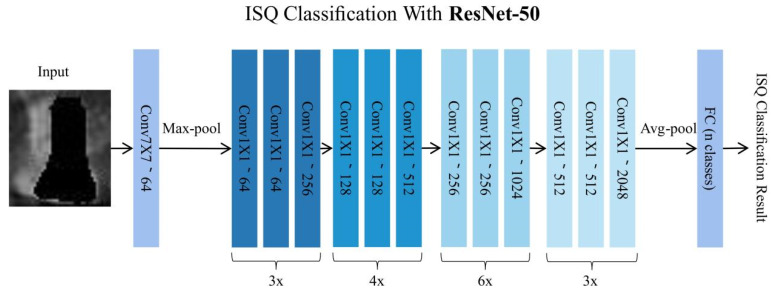
Network structure of ResNet-50.

**Figure 4 diagnostics-12-02673-f004:**
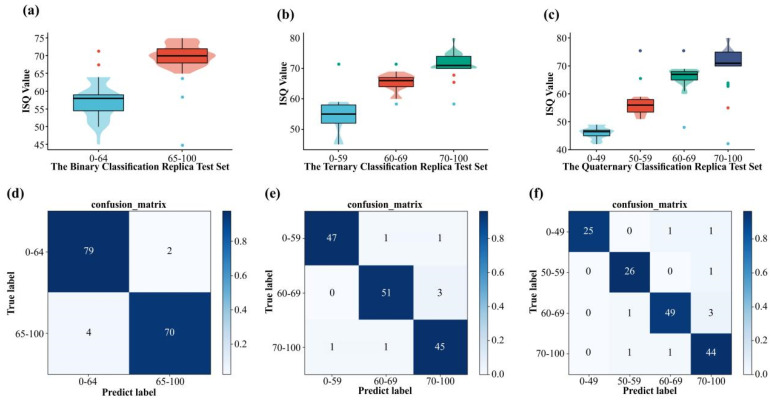
(**a**–**c**) Box plot of predicted values for each classification network; (**d**–**f**) confusion matrix for each model.

**Table 1 diagnostics-12-02673-t001:** Performance of implant identification and segmentation based on MobilenetV2-DeeplabV3+.

Performance	Mean	Implant	Background
mIoU	0.9440	0.8904	0.9976
PA	0.9676	0.9363	0.9989
Recall	0.9687	0.9383	0.9991
Precision	0.9733	0.9478	0.9987

mIoU, intersection over union; PA, pixel accuracy.

**Table 2 diagnostics-12-02673-t002:** Classification performance of implant stability based on ResNet-50 in different modes.

One-vs.-Rest Classification	Sensitivity	Specificity	PPV	NPV	F1
**Binary Classification as model outputs**					
65–100 vs. 0–64	0.9753	0.9459	0.9518	0.9722	0.9591
**Three Classification as model outputs**					
0–59 vs. others	0.9592	0.9804	0.9792	0.9804	0.9692
60–69 vs. others	0.9444	0.9792	0.9623	0.9691	0.9534
70–100 vs. others	0.9574	0.9612	0.9184	0.9802	0.9379
**Four Classification as model outputs**					
0–49 vs. others	0.9259	1.0000	1.0000	0.9844	0.9630
50–59 vs. others	0.9630	0.9920	0.9286	0.9920	0.9458
60–70 vs. others	0.9245	0.9800	0.9608	0.9608	0.9427
70–100 vs. others	0.9565	0.9533	0.8980	0.9549	0.9272

NPV, negative predictive value; PPV, positive predictive value; F1, F1 score.

## Data Availability

Te data that support the fndings of this study are available on request from the corresponding author, Linhu Ge. The raw/processed data required to reproduce these findings cannot be publicly available at this time as the data also comprise part of an ongoing study.

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
