# Peer review of "The Construction and Evaluation of a Multi-Task Convolutional Neural Network for a Cone-Beam Computed-Tomography-Based Assessment of Implant Stability"

_diagnostics, 2022, doi:10.3390/diagnostics12112673_

Round 1

Reviewer 1 Report

I should congratulate the authors for this study that assessed implant stability using artificial intelligence via convolutional neural networks . The manuscript is well written, introduction with clear rational of study and aims. Clear methodology, sample size calculation and inclusion criteria and photos. Appropriate reporting of results with comperhesive discussion. The only very minor revsion. I would be asking the author to state some definition for convolutional neural networks in the introduction and highlight the limitations and disadvantages of using of artificial intelligence. 

Author Response

Comment: State some definition for convolutional neural networks in the introduction and highlight the limitations and disadvantages of using of artificial intelligence.

Response: Thank you for your comments. We have added the definition of CNNs to the revised Introduction (lines 21 to 24). We have also acknowledged how missing data are a limitation of deep learning approaches and discussed the solutions currently being presented by the academic community in the Discussion (lines 307 to 308). Moreover, we have discussed how the "black box" nature of deep learning affects its application and adoption (lines 338 to 340), which is an issue that we hope to look into in the future.

Reviewer 2 Report

This paper assesed implant stability is integral to dental implant therapy. This study 15 aimed to construct a multi-task cascade, convolution neural network to evaluate implant stability 16 using cone-beam computed tomography (CBCT). It is clear and easy to follow.

Equation in line 183 can be removed.

Presentation of Table 1 and 2 can be improved.

Conclusion should be re-written. The first two sentences " This study is the first attempt t..... and implant stability classification." are not conclusion and must be removed. The third sentence is results and need to be revised.

Author Response

Comment 1: Equation in line 183 can be removed.

Response: Thank you for your suggestion. We have accordingly removed the equation.

Comment 2: Presentation of Table 1 and 2 can be improved.

Response: We apologise for the careless presentation. We have accordingly revised Tables 1 and 2 to improve their clarity and appearance.

Comment 3: Conclusion should be re-written. The first two sentences " This study is the first attempt t..... and implant stability classification." are not conclusion and must be removed. The third sentence is results and need to be revised.

Response: Thank you for your valuable advice. We have accordingly rewritten the conclusion to make it more concise.